# New Paradigms in Brassinosteroids, Strigolactones, Sphingolipids, and Nitric Oxide Interaction in the Control of Lateral and Adventitious Root Formation

**DOI:** 10.3390/plants12020413

**Published:** 2023-01-16

**Authors:** Maria Maddalena Altamura, Diego Piacentini, Federica Della Rovere, Laura Fattorini, Giuseppina Falasca, Camilla Betti

**Affiliations:** 1Department of Environmental Biology, Sapienza University of Rome, 00185 Rome, Italy; 2Department of Biosciences, University of Milan, 20133 Milan, Italy

**Keywords:** auxin, MAX2, nitric oxide signaling, post-embryonic root development, PUCHI, root formation in cuttings, stress response, synthetic strigolactones, 24-epibrassinolide, very long chain fatty acids (VLCFAs)

## Abstract

The root system is formed by the primary root (PR), which forms lateral roots (LRs) and, in some cases, adventitious roots (ARs), which in turn may produce their own LRs. The formation of ARs is also essential for vegetative propagation in planta and in vitro and for breeding programs. Root formation and branching is coordinated by a complex developmental network, which maximizes the plant’s ability to cope with abiotic stress. Rooting is also a response caused in a cutting by wounding and disconnection from the donor plant. Brassinosteroids (BRs) are steroid molecules perceived at the cell surface. They act as plant-growth-regulators (PGRs) and modulate plant development to provide stress tolerance. BRs and auxins control the formation of LRs and ARs. The auxin/BR interaction involves other PGRs and compounds, such as nitric oxide (NO), strigolactones (SLs), and sphingolipids (SPLs). The roles of these interactions in root formation and plasticity are still to be discovered. SLs are carotenoid derived PGRs. SLs enhance/reduce LR/AR formation depending on species and culture conditions. These PGRs possibly crosstalk with BRs. SPLs form domains with sterols within cellular membranes. Both SLs and SPLs participate in plant development and stress responses. SPLs are determinant for auxin cell-trafficking, which is essential for the formation of LRs/ARs in planta and in in vitro systems. Although little is known about the transport, trafficking, and signaling of SPLs, they seem to interact with BRs and SLs in regulating root-system growth. Here, we review the literature on BRs as modulators of LR and AR formation, as well as their crosstalk with SLs and SPLs through NO signaling. Knowledge on the control of rooting by these non-classical PGRs can help in improving crop productivity and enhancing AR-response from cuttings.

## 1. Introduction

The root system is formed by the primary root (PR), which forms LRs and, in some cases, by ARs. LRs have a post-embryonic origin and come from the PR pericycle. ARs are generally post-embryonic as well. Rarely, (e.g., in rice) they are also embryonic [1]. ARs originate from various tissues of the aerial organs and are formed by the pericycle only in the case of the hypocotyl [2]. The density of ARs and/or LRs of tap or fibrous root-systems is crucial for plant survival in altered environments [3]. The formation of ARs is also essential for vegetative propagation in planta and in vitro and for breeding programs. AR formation in cuttings is related to the abiotic stress caused by wounding and disconnection from the water- and nutrient supply of the donor plant [4]. The wound response activates a transcriptome reprogramming that promotes the expression of key factors essential for AR formation, as demonstrated in Arabidopsis leaf explants [5].

Lateral and adventitious root developmental programs share a large number of genes and phytohormone-based control mechanisms [6,7,8,9,10,11], but also show some differences in the regulation of development [12] and in the response to a specific stress [13].

Understanding the role of PGRs and gene networks in regulating formation and development of LRs and ARs is important. LR and AR formation share numerous genes. However, the mechanisms underlying the formation of ARs, and their different types and developmental stages, need further investigation. SLs and SPLs are emerging as new compounds influencing post-embryonic rooting. Their interactions need to be investigated at transcriptional and post-transcriptional levels.

Auxin is the phytohormone with a pivotal role in the control of LR and AR formation, both at the level of synthesis and transport, and is the hormone indispensable for AR formation in most in vitro cultured cuttings [14]. However, there is evidence that, in addition to and linked with auxin, BRs, SLs, and SPLs are important players in this context. Based on recent findings, BRs will be described here as modulators of rooting, able to act at the convergence between normal development and acclimation to stress. Their activity is integrated with that of SLs and possibly with signaling pathways of SPLs, which affect both stress-induced and developmental processes. ARs can be formed in response to stress, can form their own LRs, and both root types contribute to the formation of the post-embryonic root system. Therefore, knowledge about compounds not yet fully investigated in this context will be useful for improving root generation, root system development, crop fitness, and plant tolerance to various environmental stresses.

## 2. Brassinosteroids Positively Affect Lateral and Adventitious Rooting Involving an Adaptive Stress Response

BRs, steroid molecules acting as PGRs, control the geometry of the PR apical meristem in Arabidopsis [15]. BRASSINOSTEROID INSENSITIVE1 (BRI1), and paralogs, are the plasma membrane receptors of BRs [16]. BRI1 binds to brassinolide (BL), the most active form of BRs, which activates a downstream signal transduction involving numerous proteins up to the transcription factors BRI1-EMS SUPPRESSOR1 (BES1/BZR2) and BRASSINAZOLE-RESISTANT1 (BZR1). BES1 and BZR1 ultimately regulate plant growth and development [17,18]. New components in the signaling pathway and target genes of BES1 and BZR1 have been identified, indicating the existence of transcriptional networks of BR response/crosstalk with other signaling pathways [19]. For example, in the case of Arabidopsis root system, a *Tetratricopeptide-repeat Thioredoxin-like* (*TTL3*) gene, known to participate in the BR signaling pathway, has been characterized in relation to LR emergence and growth [3].

BRs play role in root system development [3,20]. Auxin and BRs interact [21,22,23,24] by having overlapping activities and sharing target genes [25]. Auxin and BRs have a common role in the organization of the PR apical meristem [26]. In Arabidopsis, BRs interact with auxin signaling to promote LR development, as also shown by many BR-deficient mutants with reduced LR formation [21]. Low levels of exogenous BRs promote LR formation by enhancing auxin transport. However, higher concentrations suppress LR formation [21,27].

BZR1/2 directly regulates the AUXIN RESPONSE FACTORS (ARFs) transcription factors involved in the transcriptional output of auxin [28]. Among these, ARF6 and ARF8 positively regulate AR formation in planta and in the thin cell layer (TCL) culture system [29]. Moreover, BIN2 regulates LR organogenesis by phosphorylating ARF7 and ARF19 [30,31]. Application of 24-epibrassinolide (eBL), an active BR [32], induces expression of several Aux/IAA genes involved in root development [33]. Application of eBL at a very low level enhanced not only LR, but also AR formation [27]. BR biosynthesis is implicated in the initiation and growth of ARs in rice. Treatments with auxin cause enhanced expression of the BR receptor gene *OsBRI1* [1].

BRs also mitigate the effects of abiotic and biotic stresses [1,34]. Root systems with numerous LRs and ARs favor anchorage, as well as water and nutrient supply from the soil/culture medium, and thus ensure plant survival. BRs play a strategic developmental/adaptive role, as these promote post-embryonic rooting even in the presence of pollutants [27]. In Arabidopsis, the quiescent centre (QC) is established at stage VII of primordium growth in both LRs and ARs (Figure 1A,B) [10,35]. Cadmium (Cd) causes QC-anomalies and auxin delocalization in both LR and AR apices starting from this stage [13], (Figure 1C,D). However, treatments with low levels of eBL (1 and 10 nM), alone (Figure 1E–H) or combined with CdSO_4_ (Cd) (Figure 1I), enhanced LR and AR formation and counteracted the Cd-induced QC/auxin anomalies [27] (Figure 1). Treatment with eBL also promoted early development of secondary roots in maize, thereby enhancing resistance to lodging and protecting the plant against stem and sheath rot [36].

Cuttings are exposed to various stress conditions during AR formation starting from the initial one, i.e., detachment from the donor plant.

In barberry cuttings, periodical water deficit or sudden temperature changes are examples of stress in culture [37]. Exogenously applied BRs helped to overcome such stresses through increased relative water content, chlorophyll concentration, photosynthetic rate, and soluble sugar content. These effects were associated with improved AR formation and elongation [37]. BRs exhibited positive effects on AR formation and growth also in cuttings of other species, e.g., geranium, Indian coleus, basil, tomato, and chrysanthemum [38,39,40], but can have also inhibitory effects, as in grapevine cuttings [41]. AR promotion is further enhanced by joint applications of BRs and auxin [37]. Exogenous BRs and auxin also favored the production of free amino acids in barberry and rhododendron cuttings [37,42]. Free amino acids are important for the regeneration of tissues subjected to stresses [43].

It is known that a coordinated, auxin-dependent, remodeling of microtubules (MTs) is involved in the induction of asymmetric cell divisions required for LR/AR initiation in planta and in in vitro culture. A cross-regulation between BRs and ethylene (ET) in controlling MT functions has been proposed [1], based on the BR role in directing cell divisions in Arabidopsis root meristems [44], and in controlling directional growth by modulating MT arrangement, which, in turn guides the positioning of cellulose microfibrils [15]. In Arabidopsis, BRs regulate the expression of the ET-induced *ERF115* gene, which encodes for a limiting factor for QC cell divisions; moreover, BR treatments enhance not only ET, but also jasmonate (JA) levels [1].

ET and JAs are PGRs involved in biotic and abiotic stress responses. When combined with auxin, they induce AR formation in planta [45] and in cultured explants, e.g., tobacco and Arabidopsis TCLs [29,46]. In Arabidopsis leaf explants, wounding rapidly induced the expression of the ETHYLENE RESPONSE FACTOR (ERF) transcription factors ERF109 and ERF111. In turn, the ERFs induced the expression of *ASA1*, a gene coding for an auxin biosynthetic enzyme. This contributed to promote rooting by providing high levels of auxin near the wounding site of the explant [47].

Further evidence for the link between stress responses and induction of rhizogenesis comes from LR formation. In fact, the biosynthesis of camalexin, a metabolite involved in stress tolerance, occurs in the LR founder cells to allow normal LR growth [48].

Collectively, results show that auxin induces LR and AR formation as adaptive response to stress by acting in combination with other PGRs, including BRs [49]. However, small molecules, e.g., nitric oxide (NO) (Figure 2), and other compounds, e.g., SLs and SPLs, up to now less investigated in the context of LR and AR formation, may be linked to BRs, as well as other canonical root inducing PGRs in the intricate control of post-embryonic rooting processes.

The relationship between BRs, SLs, and SPLs, with NO as central mediator, is discussed in the following paragraphs and summarized in Figure 3.

## 3. Nitric Oxide Is Involved in the Control of LR and AR Formation and Is Linked to the Hormonal Network Including BRs

Nitric oxide (NO) is a very important reactive nitrogen species involved in many responses to stress. However, NO is a multifunctional molecule also regulating plant developmental processes [50], including LR and AR formation [51,52,53,54,55]. It is produced in numerous organelles, including peroxisomes, but also in the cytosol [56]. Cellular NO production causes both beneficial and harmful effects, depending on its local concentration, site of synthesis, interaction with other molecules, and balance with reactive oxygen species [57]. Apart from its role in defense against pollutants [56,58,59], several studies point to NO as a key signal molecule during root system formation and development, acting at the convergence of growth and stress responses [55,56,60,61].

Moreover, the conversion of the natural auxin precursor indole-3-butyric acid (IBA) into the functional auxin indole-3-acetic acid (IAA) occurs in the peroxisomes through the β-oxidation pathway [62], also generating NO [55]. In in vitro cultured TCLs of Arabidopsis, AR-formation is totally under the control of exogenous auxins, especially IBA [63]. IBA positively affects IAA transport and the expression of IAA-biosynthetic genes, such as *Anthranilate Synthase-Alpha1* (*ASA1*) and *Anthranilate Synthase-Beta1* (*ASB1*). Consequently, AR-formation in IBA-treated TCLs obtained from the *ech2ibr10* mutant (in which the IBA-to-IAA-conversion is blocked) was highly reduced [63]. Thus, in this system, NO, the by-product of IBA-to-IAA conversion, promotes AR-formation by acting as an IAA downstream signal, as shown by epifluorescence analyses [63]. This hypothesis is also supported by the observation that, in cucumber plants, NO accumulated after IAA treatment and the NO-donor sodium nitroprusside (SNP) stimulated AR formation [64]; similar results were obtained in tomato plants [65].

By the application of NO-specific donors or under stress-conditions, it has been also proven that NO modulates auxin levels by affecting its biosynthesis, degradation, conjugation, distribution, and signaling, suggesting that NO may also function upstream of auxins [66,67]. In accordance, Arabidopsis mutants with altered NO levels also show changes in auxin biosynthetic enzyme activity, resulting in abnormal auxin levels and changes in root meristem structure [68].

Moreover, prohibitin 3 (PHB3), initially involved in stress responses, has been recently identified as a new regulator of LR initiation [69]. PHB3 causes NO accumulation, which in turn causes degradation of the AUX/IAA proteins IAA14 and IAA28. NO increases the expression of the transcription factor *GATA23*, thereby favoring cell divisions in the pericycle LR founder cells [70] in accordance with the known role of *GATA23* as the earliest marker for LR founder cell specification [71].

It is widely known that endogenous levels of NO are affected not only by auxin, but also by other PGRs, such as ET and JAs. On the other hand, NO may also affect their biosynthesis, catabolism, conjugation, transport, perception, and/or transduction [72]. For example, NO was detected at the early stages of auxin-dependent AR formation in pericycle cells of Arabidopsis hypocotyls, and its production was enhanced by exogenous JAs, but inhibited by them at stage VII of AR-primordium development [73]. As this is the stage of QC definition [10] (Figure 1A,B), this suggests that the primordium no longer depends on NO-JA when it becomes able to sustain its own growth. In some developmental processes, crosstalk between NO and JA has been reported to be mediated by peroxisomal cis-(+)-12-oxo-phytodienoic acid reductase (OPR3). OPR3 is involved in JA biosynthesis, and its expression is known to be enhanced by NO [74], thus indicating a synergism between the two. By contrast, an antagonistic relationship seems to exist between NO and ET, even if the NO-donor SNP sometimes stimulates ET production, e.g., in Arabidopsis roots subjected to Fe deficiency [75]. Moreover, in Arabidopsis, mutations in the *Ethylene Insensitive 2* (*EIN2*) gene suppress the early senescence phenotype of NO-deficient mutants, suggesting that the protein may play a key role in the crosstalk between ET and NO signaling pathways, at least under stress conditions [76]. A further role by ETHYLENE-INSENSITIVE3 (EIN3)/ETHYLENE-INSENSITIVE3-LIKE (EIL1) transcription factors, acting in the final steps of the ET signaling pathway, has been more recently suggested [77].

There is also a relationship between BRs and NO. It is, in fact, known that BRs interact with NO in numerous plant developmental processes and stress responses. For example, co-treatments with low levels of eBL (up to 1 µM) and the NO donor S-nitroso-N-acerylpenicillamine, (SNAP) enhance AR formation in *Cucumis sativus*, while co-treatments with 10 nM eBL and the NO scavenger 2-(4-carboxyphenyl)- 4,4,5,5-tetramethylimidazoline-1-oxyl3-oxide(c-PTIO) completely abolished the eBL promotion of LR formation in Arabidopsis seedlings [78,79].

Recently, we have shown that the application of SNP increased LR formation in Arabidopsis only slightly less than eBL (10 nM), while co-treatments with SNP and eBL did not result in synergistic enhancement; indeed, LR density was more similar to that obtained with SNP than with eBL alone [27]. Thus, even if the specific mechanism of BR/NO interaction still needs further study, their relationship is gradually emerging. Some evidence indicates that BRs could regulate endogenous NO levels possibly by inducing *Nitrate Reductase* (*NR*) and *NO Synthase* (*NOS*)-like genes, as occurs during LR formation in Arabidopsis [79,80]. However, there is also the inverse possibility, namely, that NO modulates BR levels. Studying the NO sensing in Arabidopsis hypocotyls, an overlapping expression pattern among genes upregulated by NO and those upregulated by BRs has been found. Additionally, a NO-induction of transcription factors leading to the expression of genes involved in BR-regulated processes has been demonstrated [81].

Our research group has been investigating the role of NO in LR/AR formation in the presence of BRs in Arabidopsis [27]. We observed the epifluorescence signal of endogenous NO in apical meristems of root primordia/elongated roots of seedlings treated with various eBL concentrations. The analysis was focused on apical meristems because they are known to be affected both by NO [58,82] and BRs (Figure 1E-H), and because they are the site of the localization of the auxin maximum, a pre-requisite for the LR/AR development in Arabidopsis, as well as other plants, e.g., rice [10,83].

Della Rovere and coworkers [27] have shown that the most effective eBL treatment (i.e., 10 nM) causes an increased NO signal in LR/AR primordia and LR/AR apices (Figure 2C–E) in comparison with the untreated controls, which show a low NO signal (Figure 2A,B). Collectively, these results suggest that exogenous BRs enhance LR/AR formation by increasing endogenous NO levels.

Taken together, all these data point to an involvement of NO in the control of LR and AR formation and suggest that this role is linked to the hormonal network, including BRs. However, further studies with multiple approaches and a wide range of species are necessary to better understand the relationship between BRs and NO.

## 4. Strigolactones Are Involved in the Regulation of Lateral and Adventitious Root Formation with a Putative NO-BR Interplay

SLs are carotenoid-derived terpenoid lactones secreted by the roots of nearly 80% of terrestrial plants [84]. The ability to synthesize SLs is a unique feature of plants [85]. SLs are implicated in plant responses to diverse abiotic stresses, such as nutrient deficiency, salinity, drought, or chilling [86,87,88,89]. The more than 30 SLs identified until now also exhibit multiple roles in regulating plant growth and development [90,91,92,93,94]. Amongst them, SLs are involved in root system architecture [84,88,95,96,97,98,99].

SL-biosynthesis takes place mainly in the root [100] and secondarily in the stem [101]. They are produced at extremely low concentrations by an isomerase (D27) and two carotenoid cleavage dioxygenases (CCDs), i.e., CCD7/MAX3 and CCD8/MAX4, which convert β-carotene to carlactone. Carlactone is, in turn, converted to carlactonoic acid by the activities of one or more cytochrome P450(s) along with oxidase(s). Carlactonoic acid is the common precursor of naturally occurring SLs [88,102]. The DWARF14 (D14) protein is the SL receptor. The SL activates D14, which in turn deactivates SL by hydrolytic degradation [103] and then binds the MORE AXILLARY GROWTH2 (MAX2/D3) F-box type protein, which assigns DWARF53 (D53) and SUPPRESSOR OF MAX2 1-like (SMXL) repressors for proteasomal degradation, resulting in the induction of gene expression [94,104].

The application of GR24 (a synthetic SL) suppresses LR and AR formation in Arabidopsis PRs and hypocotyls, respectively [95,96], but also suppresses LRs in *Medicago truncatula* [105], as well as ARs in pea shoot cuttings [95,96] and mung bean hypocotyls [106]. In Arabidopsis, the *max2* mutant produces more LRs than the wild type [107]. In mung bean hypocotyls, hydrogen peroxide and plasma membrane H⁺-ATPases seem to be the target of SL-mediated inhibition of ARs [106]. At least in Arabidopsis, treatments with GR24 do not negatively influence LR initiation, but rather the LR priming and emergence events [108]. Moreover, in tomato transgenic lines exhibiting reduced levels of CCD8, an increased number of ARs is formed in comparison with the wild type, supporting the negative role of SLs on post embryonic rooting previously observed in Arabidopsis [96,109]. However, the role of SLs is not the same in all dicots. In fact, in rapeseed, low concentrations of exogenous SLs (GR24) promoted LR formation [110]. The action of SLs on LR and AR formation seems even more complex in monocots. In fact, rice mutants with impaired SL biosynthesis and signaling exhibited reduced AR formation compared with the wild type, and the application of GR24 increased AR number, showing a positive regulation of AR formation by SLs [97]. GR24 application decreased the expression of the *DR5::GUS* system, which monitors auxin tissue localization, indicating that SLs modulate rice AR formation by negatively interfering with polar auxin transport [97].

Polar auxin transport mainly depends on the auxin efflux proteins (i.e., PINs), which create local auxin maxima to form the basis for LR and AR initiation and elongation [10,83]. In pea and Arabidopsis, SLs target processes that are dependent on the auxin flow canalization and involve auxin feedback on PIN subcellular distribution [111]. SL-deficient and SL-insensitive rice mutants formed a greater number of secondary LRs than the wild type, and GR24 application reduced this number [112]. No secondary LRs were produced in response to GR24 in the *OsPIN2* overexpressing lines although the *DR5::GUS* levels were higher than in the wild type [112]. Altogether, these results show that, in rice, SLs act by modulating local auxin levels via deregulating its efflux.

SLs interact with numerous PGRs, besides auxin, such as ET, gibberellins (GAs), cytokinins, and JAs, cumulatively or individually regulating plant morpho-physiological processes [84,110], including root architecture [98]. For example, in Arabidopsis, the GR24-induced effect on LR development depends on a close interplay of SLs with cytokinins as well as auxin [108]. In etiolated Arabidopsis, SLs increased ET biosynthesis by positively regulating the transcription of the *ACO* genes, which oxidize 1-aminocyclopropane-1-carboxylic acid (ACC) to ET, without affecting the stability of the ACC biosynthetic gene *ACS* [113]. SLs also interact with GAs, key regulators, and molecular clocks for root meristem development [114]. In fact, D14, a member of the family of α/β-fold hydrolases of the SL pathway, is also a GA receptor [115,116].

The literature data indicate a crosstalk between SL and BR signaling pathways. In fact, MAX2, a component of SL signaling, suppresses shoot branching by interacting with BZR1 and BES1, key regulators of BR-dependent gene expression, with this leading to the degradation of these transcription factors [117]. Moreover, during the same process in rice, SLs and BRs antagonistically regulate the stability of the complex formed by D53 of the SL signaling pathway and OsBZR1 of the BR signaling pathway [118]. Furthermore, a transcriptome analysis pointed to BES1 as a transducer of SL effects on drought memory [119]. Advances in the regulation of SLs and BRs pathways are numerous [19]; however, their interaction still needs investigation.

NO and SLs are common regulatory signals in stress responses and root formation. NO enhances salinity tolerance in tomato seedlings by enhancing SL synthesis [89]. A SL–NO interplay exists during root formation in sunflower [120], maize [121], rice [87], and Arabidopsis [88]. For example, in sunflower seedlings, LR and AR formation are reduced by GR24 treatments, which also caused decreased NO levels in LR apices, as well as increased levels in the PR apex [120]. In addition, a strong increment in the activity of the SL biosynthetic enzyme CCD occurs in the presence of the NO scavenger cPTIO. Taken together, these results indicate a negative regulatory effect of endogenous NO on SL biosynthesis [120]. In the PR apices of nitrate-starved maize seedlings, short-term nitrate treatments repressed the transcription of genes involved in SL biosynthesis and transport, and cPTIO enhanced the transcription of SL biosynthetic genes *CCDs* [121]. Even if in these studies NO seems to act as a negative regulator of SL signaling, the nature of the NO–SL relationship seems more complex, because NO can also be a positive regulator of SLs. In fact, by using rice NO-deficient mutants (*nia2-1*, *nia2-2*) grown under phosphate and nitrogen deficiency, it has been demonstrated that both NO and SLs are positive regulators of AR meristem activity and of consequent root elongation [87,122]. In a recent investigation on the SL–NO interaction in the root system of Arabidopsis under unstressed conditions, it has been shown that *Atmax1-1* and *Atmax2-1* mutants are deficient in SL synthesis and signaling, respectively. They also exhibited increased NO levels compared to the wild type, with the increases being due to decreased S-nitrosoglutathione (GSNO) reductase (GSNOR) protein abundance/activity. The downregulation of SL biosynthetic genes (*CCD7*, *CCD8*, and *MAX1*) in *gsnor1-3*, a GSNOR-deficient mutant, and its pronounced sensitivity to exogenous SL, together with the insensitivity of *Atmax1-1* and *Atmax2-1* mutants to exogenous GSNO, suggest that a functional GSNOR is needed to control NO levels for SLs to function in root growth [88].

Considering the existence of a crosstalk between SL and BR signaling, e.g., in shoot branching [117,118], and hypocotyl elongation [81], and that of a NO-SL interaction, there is a consistent possibility that NO may be the central node linking the SL and BR pathways in the control of post-embryonic rooting. This proposed model is shown in Figure 3.

## 5. Role of Sphingolipids in Rooting with a Putative NO–BR Interplay

SPLs are abundant in the plasma membrane, but also in endomembranes. Plant SPLs build raft domains with membrane sterols. Functional genomics studies have shown that various SPLs are involved in plant growth and development, including post-embryonic rooting, as well as in stress responses [123,124,125,126,127,128,129]. By using lipidomic techniques, over 300 plant SPLs have been isolated, which are divided in four major classes: long-chain bases (LCB), ceramides (Cer), glucosylceramides (GluCer), and more complex glycosylated sphingolipids, known as glycosyl-inositolphosphoryl-ceramides (GIPC) [128].

Fatty acids with 22 or more carbon atoms are known as very long chain fatty acids (VLCFAs); they are synthesized in the endoplasmic reticulum and are the precursors for the synthesis of SPLs. In the endoplasmic reticulum, a long-chain base (LCB) SPL is generated by L-serine condensed with palmitoyl-CoA, which is reduced, and N-acylated, to form ceramide (Cer). Ceramide Synthases (CerS), encoded by the multigenic family of *Lag One Homolog* (*LOH*), are responsible for the formation of the amide bond between VLCFA and LCB, leading to Cers. Cers are substrates for complex SPLs, including inositol phosphorylceramide (IPC), and GluCer, produced in the Golgi apparatus. In addition to hydroxylation, Cers and LCBs can be phosphorylated to give a variety of active molecules [128,130,131].

An interplay between VLCFAs, Cer and LCB signals seems essential for stress signaling [128]. In addition, GIPCs, one of the two major SPL classes in the plasma membrane, are required for normal growth and reproduction in Arabidopsis [129]. Functional analysis of the Cers family in Arabidopsis demonstrates that very-long-acyl-chain (C > 18 carbons), but not long-chain SPLs, are essential for post-embryonic rooting. In fact, in Arabidopisis, the reduction of very-long-chain fatty acid SPLs levels leads to auxin-dependent inhibition of LR emergence, which is associated with selective aggregation of the plasma membrane auxin carriers AUX1 and PIN1 in the cytosol [132]. Phosphorylation status may be important for the SPL response. In fact, Cers and free LCBs cause the hypersensitive response and programmed cell death (PCD) in plants, whereas their phosphorylated forms show the opposite effect [133,134,135].

SPLs are targeted by phytohormones in a cell- or tissue-specific manner to regulate plant development by regulating other signaling pathways through feedback mechanisms [136]. Positive roles of very long chain SPLs have been demonstrated in cytokinesis and auxin transport [137,138,139]. Auxin and SPLs are involved in LR formation. The formation of LR primordia in Arabidopsis needs a pericycle–endodermal communication through VLCFAs, and the condensing enzymes KCS2, KCS20, and KCS1 are expressed in the PR endodermis [140,141]. The auxin-regulated AP2/ERF transcription factor PUCHI controls both early stages of LR primordium formation [142], as well as the distribution of hormonal signals during late LR organogenesis [143]. PUCHI targets the spatial expression of the genes involved in VLCFA synthesis during LR development [144]. VLCFAs, in turn, repress cell divisions at the flanks of the LR primordium and regulate the activity of AUX1/PIN1 auxin carriers, thus collaborating with the establishment of the auxin maximum at stage VII of LR primordium development [136,144]. In accordance, VLCFA-deficient mutants display defects in LR formation, and the lipidomic profiling of *puchi-1* mutant roots revealed an altered VLCFA content with a reduction of C24 SPLs [144]. In addition, the auxin-induced mitogen-activated protein kinase MPK14 promotes LR development by alleviating the repression of VLCFA biosynthesis mediated by the AP2/ERF transcription factor ERF13 [145,146].

The observation that SPL-specific VLCFAs are the targets of auxin-PUCHI mediated LR formation is consistent with the role of the Cer-synthesis LOH in the same process. In fact, in the *loh* mutant, both AUX1 and PIN1carriers are mislocalized in endosomal compartments [136]. Since both AUX1 and PIN1 are necessary, not only for LR formation, but also for AR formation [10], the control by SPL-specific VLCFAs can likely be extended to adventitious rooting in planta and in vitro. Indeed, regeneration of ARs in vitro usually starts with the induction of a pluripotent callus through an indirect process on an auxin-rich medium (CIM). In Arabidopsis, it has been demonstrated that CIM-induced callus formation occurs from the pericycle or pericycle-like cells, following a root developmental pathway involving VLCFAs [141].

SPLs also interact with ABA, JA, and ET [131,135,147].

The relationship between SPLs and SLs has been poorly investigated. However, it is important to note that VLCFAs not only are the precursors of SPLs [148], but their production is also induced by SLs, e.g., during cotton fiber elongation [149]. In this process, the expression levels of four *KCS* genes coding for condensing enzymes for long (>C20) fatty acid elongation are upregulated by treatments with GR24, and consequently the levels of C22:0 and C26:0 VLCFAs increase [149].

Information about the relationship between SPLs and BRs in plant morphogenesis is still very limited. During early olive fruit development, a BR-induced modulation of long-chain SPLs composition and gene expression was found [150]. The application of eBL reduced the content of LCBs and the transcript levels of some SPL-related genes. However, application of the BR-biosynthesis inhibitor brassinazole increased LCB content and the related gene expression, suggesting that BRs might negatively regulate the contents of long-chain SPLs during olive fruit development, alleviating their negative effects [80,150]. In plant immunity, a coordination of BRs with SPLs (LCBs) has been recently hypothesized, with LCB accumulation affecting BR signaling [151]. In accordance, hydroxyl groups of SPL chains affect the abundance of the transmembrane co-receptor BRASSINOSTEROID INSENSITIVE 1 -ASSOCIATED RECEPTOR KINASE 1 (BAK1), involved in controlling the early events of the BR-signaling pathway [152]. In Arabidopsis, where 2-hydroxy SPLs are necessary for the organization of plasma membrane nanodomains, these compounds affect the abundance of BAK1, demonstrating the existence of an interaction between SPL and BR signaling in plant immunity [153]. It has been very recently suggested that BRs affect the specific orientation of unsaturated/saturated fatty acids in the membranes. In turn, fatty acid organization favors the construction of membrane areas enriched in BRs, with this affecting membrane responses to temperature changes [154]. We can speculate that, through this interaction between SPLs and BRs, plants also fine-tune rhizogenesis. In fact, the BR receptor BRI1 is not only regulated by plasma membrane lipids, but also by BRs themselves [136]. Thus, SPLs, by enhancing BR nanodomains in the plasma membrane, may play a role in BR-mediated root formation.

In *Taxus* sp. cell cultures, a fungus-produced SPL induced rapid and dose-dependent NO production [131]. In cold-treated Arabidopsis plants, early production of NO down-regulated the synthesis of SPLs [155]. This suppression was not observed in the *nia1/nia2* nitrate reductase mutant, which was impaired in NO biosynthesis [155]. Taken together, these results suggest that NO is involved in SPL-mediated stress responses and that a NO-SPL interaction may also be active in rhizogenesis.

Based on these data, and on the information discussed in Paragraph 3 regarding the NO interplay with BRs during LR/AR formation, we propose a model in which NO acts as modulator of the BR-SPL interaction (Figure 3).

## 6. Conclusions

On the basis of results discussed in this review, the scenario that emerges ascribes an important role to the positive/negative interactions among BRs, SLs, and SPLs in the control of LR and AR initiation and growth. While there is considerable information on BRs and SLs, the PGR-like role of SPLs is very limited. In particular, the possible relevance of carbon chain length of SPLs is yet to be explored.

Moreover, the number of plant species analyzed so far is still too low to allow generalization. Nonetheless, there is a fair amount of evidence in support of NO as a common signaling molecule for BRs, SLs, and SPLs. Thus, for the first time, and after more than 40 years of research on NO, the hypothesis is put forward here that this signaling molecule is the common interactor in the control of post-embryonic rooting by BRs, SLs and SPLs, concomitant with the role of auxin in this process (Figure 3). A great deal of research remains to be carried out to confirm this hypothesis and to characterize all the steps of the interactive pathway(s).

**Figure 3 plants-12-00413-f003:**
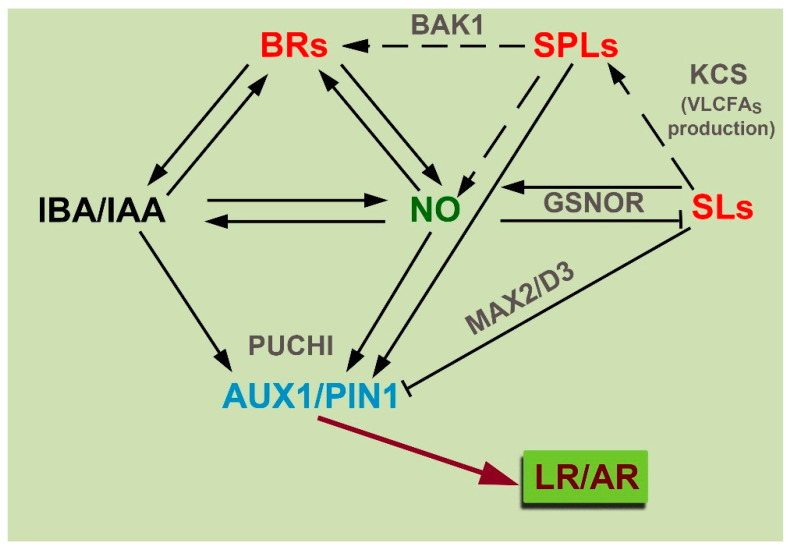
Model explaining the hypothesis of a central role of nitric oxide (NO) in the modulation of promotive/antagonistic actions of brassinosteroids (BRs), strigolactones (SLs), and sphingolipids (SPLs) in the control of the auxin-induced lateral and adventitious (LR/AR) root formation. The double arrow between BRs and NO indicates how the two compounds mutually influence their levels and activity. Indole-3-acetic acid (IAA), and its natural precursor indole-3-butyric acid (IBA), cause LR and AR formation through the activity of the auxin influx (e.g., AUX1) and efflux (e.g., PIN1) cellular carriers. NO is a by-product of IBA to IAA conversion and positively modulates auxin levels at various steps, including auxin synthesis and transport by AUX1/PIN1 carriers (arrows in Figure). SLs induce NO, which in turn negatively affects their levels. Thus, NO modulates SLs through a negative interaction. GSNOR (S-nitrosoglutathione reductase) is needed to control NO levels to allow SL functioning in LR/AR formation [88]. Moreover, SLs directly inhibit auxin efflux by PIN, thus inhibiting LR/AR formation (arrow), with a role for MAX2/D3 (MORE AXILLARY GROWTH2) F-box type protein [107,112]. Even if not shown in the Figure, a positive relationship between NO and SLs has been also reported (see the text). SLs may also induce the production of VLCFAs (very long-chain fatty acids) components of sphingolipids (SPLs) through the activities of *KCS* (*ketoacyl-CoA synthase*) genes, as in other systems [149] (dashed arrow). The accumulation of SPLs, possibly those containing VLCFAs, might positively affect BR signaling (dashed arrow) through BAK1 (BRASSINOSTEROID INSENSITIVE 1 -ASSOCIATED RECEPTOR KINASE 1 [152], but this hypothesis needs to be confirmed. PUCHI, an auxin regulated AP2/ERF transcription factor, might target the spatial expression of SPL-VLCFA genes [144], which regulate the activity of AUX1/PIN1 auxin carriers [136,144]. In addition, SPLs might positively affect NO (dashed arrow), essential to LR/AR formation. (See the text for further explanations).

## 7. Perspectives for Adventitious Rooting of Cuttings and Agriculture

Adventitious root formation is crucial for the successful vegetative propagation of plant species, necessary for agriculture, horticulture, forestry, and medicine, as well as for biotechnological applications, e.g., for the low-cost extraction of secondary metabolites, genetic amendment, and gene/genome editing. The process is essentially required for clonal propagation and conservation of elite, rare, and/or threatened germplasms [156]. Adventitious rooting occurs through a complex series of events, affected by numerous variables, both internal and external to the explant, and with specific, often antagonistic, requirements. The competent cells must change their fate to form de novo a root [2]. Thus, the inception of a new meristem by these cells may be interpreted as a combined stress/developmental response [4]. In addition, cell reprogramming must be coordinated with a correct reallocation of plant resources within the cutting [157]. This is necessary in order to channel cells towards the acquisition of the AR identity. It is not easy to reprogram development and acquire new organ-identity, and, in fact, many species are recalcitrant to production of ARs. In the face of climate change, increasing soil salinity, and widespread soil pollution, there is a need to enhance the number of stress-tolerant genotypes, useful for agriculture, the environment, and the bioeconomy. Propagation by rooted cuttings of such plants is a useful approach for producing clonal specimens on a large scale and at a low cost. Endogenous/exogenous factors, phytohormones, mainly auxins, alone or combined with other phytohormones, and expression of specific regulatory genes, have been recognized as fundamental factors for the AR-process [14]. However, the need to explore new AR-inductive compounds to increase rooting, especially in recalcitrant genotypes, and improve plant fitness, productivity, and resistance to adverse environmental conditions, remains. Another important event in AR-formation from cuttings is priming of the competent cells for initiating the new morphogenic event. Cytoskeletal changes are important for priming, and phragmoplast activity for directing cytokinesis. Three classes of compounds with important interactive roles in LR/AR formation in planta, i.e., BRs, SLs, and SPLs, discussed in this review, are also known to be involved in cytoskeletal organization during LR/AR formation [1,15,44,137,138,139]. Early cytoskeletal modulation is still a black box for adventitious rooting in cuttings and advances in knowledge will be possible using these compounds. At present, information about the application of BRs, SLs, and SPLs for rooting of cuttings of economically important plants are limited. However, BRs have been used with success for the proliferation and vegetative growth of potato [158], *Cymbidium elegans* [159], *Arachis hypogaea* [160], and a few other cases.

Available results for SLs are not encouraging [96,157]. Moreover, evidence for the use of SPLs in adventitious rooting of cuttings is almost completely absent, even though in in vitro cultured Arabidopsis explants they were shown to act as signals for mediating callus competence and root regeneration [141]. Regarding SPLs, promising results come from research on plant immunity, a process in which the use of commercial SPLs and SPL metabolism inhibitors has contributed to reveal their multiple positive functions [151].

By contrast, the field of application for BRs in agriculture is broad. Spray applications of exogenous BRs in field-grown crops have been in use for many years. SLs, instead, are only beginning to be explored as agrochemicals, and, in addition to GR24, other more easily synthesized analogs have been developed and positively tested to give higher yields [161]. Moreover, the application of SLs has been suggested to be very promising in improving drought tolerance, increasing yields, and controlling parasitic weeds [161].

## Figures and Tables

**Figure 1 plants-12-00413-f001:**
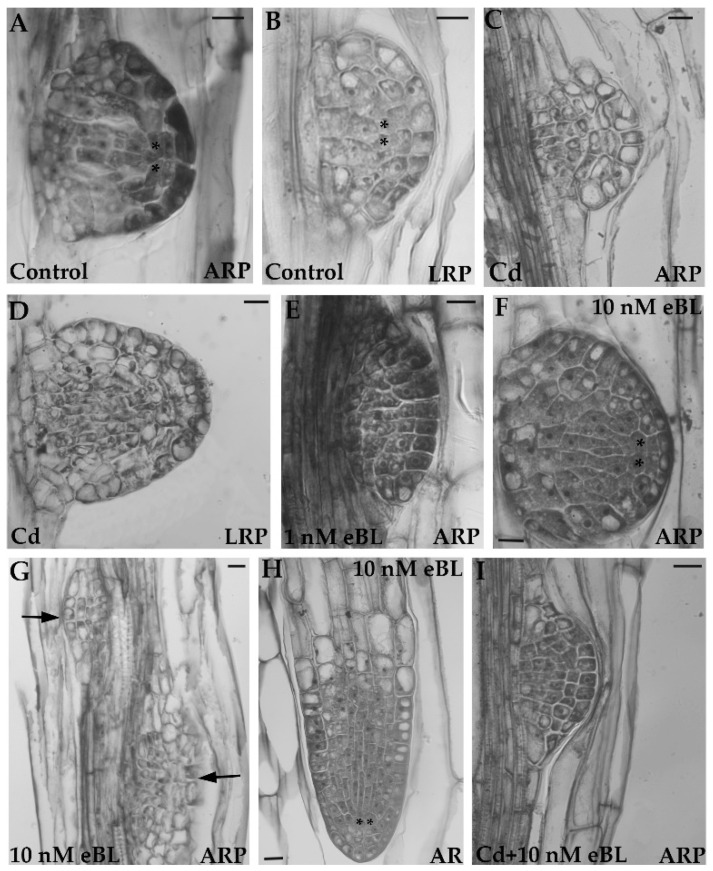
Histological images of adventitious root primordia (ARP, (**A**,**C**,**E**–**G**,**I**)) lateral root primordia (LRP, (**B**,**D**)) and apices of adventitious root (AR, (**H**)) formed following either the anticlinal proliferation of the hypocotyl pericycle (ARPs) or the anticlinal proliferation of the primary root pericycle (LRPs) in *Arabidopsis thaliana* seedlings (Col ecotype) cultured for nine days with continuous darkness followed by seven days under 16 h light/8 h darkness photoperiod, according to [27]. (**A**,**B**) regular ARP (**A**) and regular LRP (**B**) at stage VII of development, formed on the control medium, i.e., with no exogenous phytohormone (Control). The regular quiescent center (QC) presence is shown by the asterisks. (**C**,**D**), ARP (**C**) and LRP (**D**) at stage VII or soon after, showing irregular apical structure under the treatment with 60 µM of CdSO_4_ (Cd). (**E**,**G**) ARPs at different developmental stages showing regular QC definition at stage VII (**F**). Treatments with either 1 nM of 24-epibrassinolide (eBL) (**E**) or 10 nM eBL (**F**,**G**). Arrows in (**G**) show ARPs located very near each other. The image is representative of the high density of roots obtained with 10 nM eBL [27]. (**H**) AR apex with regular QC (showed by asterisks) under 10 nM eBL. (**I**) young regular ARP (stage of dome definition, [8]) formed in the combined presence of Cd and 10 nM eBL. Longitudinal radial sections stained with toluidine blue and presented in black and white. Data from Della Rovere et al. [27]. Bars = 10 µm.

**Figure 2 plants-12-00413-f002:**
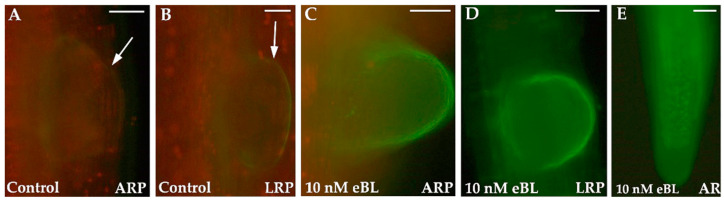
Adventitious root primordia (ARPs) (**A**,**C**), lateral root primordia (LRPs) (**B**,**D**), and apices of adventitious roots (ARs) (**E**) of Arabidopsis thaliana seedlings grown for 16 days in the absence (Control, (**A**,**B**)) or in the presence of 10 nM 24-epibrassinolide (eBL) (**C**–**E**) and then treated with the NO-specific probe 4-Amino-5-Methylamino-2’,7’-Difluorofluorescein Diacetate (DAF-FM DA). The fluorescence signal was visualized using a DMRB microscope (Leica, Wetzlar, Germany). Arrows show the inception of the dome of an ARP (**A**) and of an LRP (**B**). Bars: (**A**,**C**–**E**) = 40 µm; (**B**) = 20 µm. Data from Della Rovere et al. [27].

## Data Availability

Not applicable.

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
