# Peer review of "New Paradigms in Brassinosteroids, Strigolactones, Sphingolipids, and Nitric Oxide Interaction in the Control of Lateral and Adventitious Root Formation"

_plants, 2023, doi:10.3390/plants12020413_

Round 1
Reviewer 1 Report
This review explored the interactions of BRs, SLs and SPLs, and their relationships with other phytohormones and small molecules such as nitric oxide (NO), in the LR and AR formation processes. The study summarized many important findings. However, the content is too redundant, the summary and refinement are not in place, and many sentences in the article lack references. I wrote some my suggestions about this article.
1. Review articles should not be a list of references, but need to be summarized, refined and merged. For example, reference 103 in the fourth part of the article is not allowed to directly describe the results of others without processing.
2. Keywords - do not repeat words from the title. This way you can increase the visibility of your research, this way there are more different words that can be used to find the manuscript.
3. The third line "SL-biosynthesis takes place in roots and shoots" in the fourth part of the article is lack of references and needs to be revised according to the full text.
4. This review is mainly concerned with root development. Please reduce the response of these hormones and compounds to stress in this article.
Author Response
Reviewer 1
You wrote:
Review articles should not be a list of references, but need to be summarized, refined and merged. For example, reference 103 in the fourth part of the article is not allowed to directly describe the results of others without processing.
Answer
References were described in relation to their contents in the revised version.
For example, the reference n.103 of the old version, now reference n. 122 is cited in relation with the paper’s content and put together with ref. n. 87 (lines 346-349 of the revised version)
You wrote:
Keywords - do not repeat words from the title. This way you can increase the visibility of your research, this way there are more different words that can be used to find the manuscript.
Answer:
Key words were changed according to your suggestion
You wrote:
The third line "SL-biosynthesis takes place in roots and shoots" in the fourth part of the article is lack of references and needs to be revised according to the full text.
Answer:
Sorry, it was an oversight. The two references (100 and 101) were added in the revised version (line 277 of the revised version)
You wrote:
This review is mainly concerned with root development. Please reduce the response of these hormones and compounds to stress in this article.
Answer:
We believe that rooting can also be considered as a response to stress, in particular adventitious rooting from cuttings (see lines 15-16, 43-46, 153-154, 156-157). For this reason, and because the Editor also agrees with our approach, the references to the stress responses remained in the revised version but were slightly reduced in the revised version to follow your suggestion.
Your Comment:
You asked an extensive editing of English language and style
Answer:
The English and the style of the revised version were corrected by an English-speaking colleague of the University of Bologna (Italy) who we mentioned in the thanks (lines 554-555).
Reviewer 2 Report
In this review, the authors reviewed the research progress of brassinosteroids, strigolactones, sphingolipids, and nitric oxide, and their crosstalk in the formation of post-embryonic lateral roots and adventitious roots. Particularly, both strigolactones and sphingolipids may play important roles in root growth and development though the reports are very limited so far. Therefore, the review in this field will certainly promote the research in this field. I think this manuscript is suitable for publication in the plants after minor revision.
1. The mutant name should be in italics.
2. Abbreviations should always be used after they appear for the first time and are noted.
3. As the authors described, the NO played a central role in the network of interaction among BRs, SLs, and SPLs on the root system growth and development. Therefore, the manuscript title should be included the NO.
4. The ROS also play important roles in the functions of BRs, SPLs, and SLs on root development. It is better to discuss the ROS role in the MS.
5. Figure 3 should be more detailed, including some important genes.
Author Response
Reviewer 2
Comment n.1
The mutant name should be in italics.
Answer:
Sorry for the mistake. Mutant names are in italics in the revised version of the review.
Comment n.2
Abbreviations should always be used after they appear for the first time and are noted.
Answer:
We followed your suggestion in the revised version
Comment n.3
As the authors described, the NO played a central role in the network of interaction among BRs, SLs, and SPLs on the root system growth and development. Therefore, the manuscript title should be included the NO.
Answer:
Nitric oxide (NO) was added in the title, as suggested (line 2 of the revised version).
Comment n.4
The ROS also play important roles in the functions of BRs, SPLs, and SLs on root development. It is better to discuss the ROS role in the MS.
Answer
Our review has another target (NO and its relationship with BRs,SLs and SPLs),. In our opinion, to mention all the relationships of BRs, SLs and SPLs with ROS would make the review extremely longer and no longer focused on our main focus (NO) weaking its importance. The Editor has given us the option not to talk about it. However ROS were mentioned for the balance with NO (ref. 57) (lines 176-177) and in relation with SLs (ref 106) (lines 291-293).
Comment n. 5.
Figure 3 should be more detailed, including some important genes.
Answer
As requested, proteins/transcription factors essential for the relationship among the compounds analyzed were added in the new Fig.3 and described in its new legend, where also the related references were added.
Many thanks for your suggestions.
Reviewer 3 Report
I have gone through the text of manuscript of this review paper. There need for revising the manuscript as the present form is verbose. The revised manuscript should be evaluated by an expert of English language. I have tried restructuring the abstract, introduction, some other sections and conclusion ( kindly see enclosure). I suggest that the authors study other papers/reviews to update the information on subject ( please see enclosure).
Please complete reference 3, that is
Xin, P.; Schier, J.; Šefrnová, Y.; Kulich, I.; Dubrovsky, J.G.; Vielle-Calzada, J.-P.; Soukup, A. The Arabidopsis TETRATRICOPEPTIDE-REPEAT THIOREDOXIN-LIKE (TTL) family members are involved in root system formation via their interaction with cytoskeleton and cell wall remodelling. The Plant Journal, 112/4, November 2022, Pages 946-965
rewritten and revised paper may be accepted for publication.

Author Response
Reviewer 3
Comment:
Please complete reference n.3
Answer
The new ref 3 is now complete (lines 564-566).
Answers to your attached file
Abstract was rewritten according to your suggestions
Introduction was rewritten according to your suggestions and those of the Editor
Chapter 2-Brassinosteroids was rewritten according to your suggestions and those of the Editor
Conclusions were rewritten according to your suggestions.
Chapter 7-Perspectives was rewritten according to your suggestions.
Reviewer’s Comments in the attached file:
I suggest that the authors consult the following paper before rewriting the text of revised manuscript.
Answer:
We read all the papers in your list, and the following 12 papers were enclosed and discussed in the revised version.
Lv, B., Wei K., Hu K., Tian T., Zhang F., Yu Z., Zhang D., Su Y., Sang Y., Zhang X., and Z. Ding (2021) MPK14- mediated auxin signaling controls lateral root development via ERF13-regulated very-long-chain fatty acid biosynthesis. Mol. Plant. 14, 285–297. N. 145 of the revised version
Primc, A.; Maizel, A. Understanding lateral root formation, one cell at a time. Mol Plant. 2021, 14, 1229-1231. N. 48 of the revised version
Bellande , K., Duy-Chi Trinh , Anne-Alicia Gonzalez , Emeric Dubois , Anne-Sophie Petitot , Mikaël Lucas , Antony Champion , Pascal Gantet , Laurent Laplaze , and Soazig Guyomarc’h (2022) PUCHI represses early meristem formation in developing lateral roots of Arabidopsis thaliana. Journal of Experimental Botany, Vol. 73/11 pp. 3496–3510. N. 143 of the revised version
Bhoi, A., Bhumika Yadu, Jipsi Chandra and S. Keshavkant (2021) Contribution of strigolactone in plant physiology, hormonal interaction and abiotic stresses. Planta, 254 (2); 28 N. 100 of the revised version
Chang, X.Y., Zhang, K., Yuan, Y., Ni, P., Ma, J., Liu, H., Gong, S., yang, G. S. and M. Bai (2022) A simple, rapid, and quantifiable system for studying adventitious root formation in grapevine. Plant Growth Regulation 98/1,117-126 N. 41 of the revised version
Guyomarc’h, S., Boutte, Y. and L. Laplaze (2021) AP2/ERF transcription factors orchestrate very long chain fatty acid biosynthesis during Arabidopsis lateral root development. Molecular Plant 14, 205–207 N. 146 of the revised version
Liu,W.,Yuyun Zhang, Xing Fang, Sorrel Tran, Ning Zhai, Zhengfei Yang, Fu Guo, Lyuqin Chen, Jie Yu, Madalene S. Ison, Teng Zhang, Lijun Sun, Hongwu Bian, Yijing Zhang, Li Yang and Lin Xu (2022) Transcriptional landscapes of de novo root regeneration from detached Arabidopsis leaves revealed by time-lapse and single-cell RNA sequencing analyses. Plant Communications 3, 100306, July 11 2022 N. 5 of the revised version
Luo, L., Xie, Y. and Wei Xuan (2022) Prohibitin 3 gives birth to a new lateral root primordium. Journal of Experimental Botany, Vol. 73, No. 12 pp. 3828–3830. N. 70 of the revised version
Omoarelojie, L.O., Kulkarni, M.G., Finnie, J.F., and J. van Staden (2021) Strigolactone inhibits hydrogen peroxide and plasma membrane H+-ATPase activities to downregulate adventitious root formation in mung bean hypocotyls. Plant Growth Regulation, 94/1 p. 11-21 N. 106 of the revised version
Temmerman, A. , Belen Marquez-Garcia , Stephen Depuydt , Silvia Bruznican , Carolien De Cuyper, Annick De Keyser, François-Didier Boyer , Danny Vereecke , Sylwia Struk and Sofie Goormachtig (2022) MAX2-dependent competence for callus formation and shoot regeneration from Arabidopsis thaliana root explants. Journal of Experimental Botany, Vol. 73, No. 18 pp. 6272–6291 N. 107 of the revised version
Tian, J., Qian Xing, Tingting Jing, Xing Fan, Qingzhu Zhang & Ralf MüllerXing (2022). The epigenetic regulator ULTRAPETALA1 suppresses denovo root regeneration from Arabidopsis leaf explants, Plant Signaling & Behavior, DOI: 10.1080/15592324.2022.2031784 N. 47 of the revised version
Vangheluwe, A. and Tom Beeckman (2022) Lateral Root Initiation and the Analysis of Gene Function Using Genome Editing with CRISPR in Arabidopsis. Genes 2021, 12, 884. N. 71 of the revised version
Thanks a lot for all your suggestions.
Round 2
Reviewer 1 Report
I have no more questions and it could be acceptable for your revision.
Author Response
Thanks for your revision, Altamura
Reviewer 3 Report
The manuscript of paper may be accepted for publication.
Author Response
Thanks for your revision, Altamura